# Lower Ricci Curvature for Efficient Community Detection

**Yun Jin Park**                                                                  *yjinpark@unc.edu*
*Department of Biostatistics*
*The University of North Carolina at Chapel Hill*

**Didong Li**                                                                     *didongli@unc.edu*
*Department of Biostatistics*
*The University of North Carolina at Chapel Hill*

**Reviewed on OpenReview:** *https://openreview.net/forum?id=EoiuRII7MQ*

## Abstract

This study introduces the Lower Ricci Curvature (LRC), a novel, scalable, and scale-free discrete curvature designed to enhance community detection in networks. Addressing the computational challenges posed by existing curvature-based methods, LRC offers a streamlined approach with linear computational complexity, which makes it well suited for large-scale network analysis. We further develop an LRC-based preprocessing method that effectively augments popular community detection algorithms. Through applications on multiple real-world datasets, including the NCAA football league network, the DBLP collaboration network, the Amazon product co-purchasing network, and the YouTube social network, we demonstrate the efficacy of our method in significantly improving the performance of various community detection algorithms.

## 1 Introduction

In the modern era, the ubiquity of networks in various domains, from biological pathways (Koutrouli et al., 2020) and social networks (Ji et al., 2022) to technological and cosmic networks (De Regt et al., 2018), has fostered significant interest in the study of complex systems. These networks, characterized by nodes representing entities and edges denoting interactions, provide a framework for understanding the intricate relationships and dynamics within these systems. Graph theory, applied to these network representations, has emerged as a vital tool for dissecting and interpreting the structural and functional intricacies of these interconnected systems (West et al., 2001).

Community detection is one of the most important aspects in the analysis of complex networks (Dey et al., 2022). In these networks, communities represent subgroups of nodes (such as individuals, biological entities, or devices) that are more densely interconnected among themselves than with the rest of the network. The identification of these communities can yield invaluable insights into the structure and dynamics of the system being studied. For instance, in social networks, communities can represent groups of people with shared interests or connections, revealing patterns in social interactions and relationships (Bakhthemmat & Izadi, 2021). In biological networks, such as those representing metabolic or protein-protein interaction pathways, community detection can help identify functional modules or clusters of interacting molecules, crucial for understanding biological processes and disease mechanisms (Tripathi et al., 2019). Similarly, in technological networks, such as the internet or telecommunications networks, communities might consist of densely interconnected nodes or hubs that are critical for network functionality and resilience (Zhang et al., 2022). By discerning these communities, we can gain a deeper understanding of not only the individual elements within the network, but also the overarching principles that govern their interactions and collective behavior.

Community detection methods have evolved significantly to address the diverse and complex structures of modern networks. Hierarchical Clustering (Fortunato, 2010; Hastie et al., 2009), for instance, has been

instrumental in identifying nested community structures by iteratively merging or dividing groups based on their similarity. The Girvan-Newman algorithm (Newman, 2004; 2006), notable for its edge-betweenness centrality approach, has contributed substantially to understanding modularity within networks. Similarly, Label Propagation algorithms (Raghavan et al., 2007), recognized for their simplicity and speed, have been effective in detecting community structures in large networks by allowing nodes to adopt the majority label of their neighbors. The Walktrap algorithm (Pons & Latapy, 2005) has gained recognition for its approach of using random walks to identify communities, based on the idea that short random walks tend to stay within the same community. This method is particularly adept at capturing the local community structure in large networks. The Leiden algorithm (Traag et al., 2019), an improvement over the well-known Louvain method (Blondel et al., 2008), offers enhanced accuracy and resolution in detecting communities. It addresses some of the limitations of previous methods by refining the community boundaries and ensuring a more balanced distribution of community sizes. These methods, each with their unique approaches and strengths, have collectively advanced our understanding of network structures, contributing to fields ranging from sociological studies to biological network analysis.

A recent and significant development in network analysis is the discovery of a correlation between discrete curvature and community detection (Sia et al., 2019), which underscored the potential of using curvature-based methods to enhance our understanding and identification of communities within complex networks. Network curvature, particularly discrete curvature, has emerged as a powerful tool in the realm of graph theory and network analysis. The concept, rooted in geometric analysis, involves adapting the notion of Ricci curvature (Ricci & Levi-Civita, 1900; Do Carmo & Flaherty Francis, 1992), traditionally applied to smooth manifolds, to discrete networks. This adaptation has led to the development of various discrete curvature measures, each offering unique insights into network properties.

One of the key forms of discrete curvature is the Ollivier-Ricci Curvature (ORC, Ollivier (2007)), which has been instrumental in studying transport efficiency and robustness in networks. It provides a measure of how the network deviates from a geometrically flat structure, offering insights into network connectivity and resilience. Another significant variant is the Forman Ricci Curvature (FRC, Forman (2003)), adapted from Riemannian geometry, which has been applied to analyze the shape and topological features of networks, proving useful in understanding the underlying structure of complex systems. The Balanced Forman Curvature (BFC, Topping et al. (2021)), a refined version of the FRC, has been particularly effective in identifying bottleneck structures and critical connections within networks. This form of curvature is beneficial in applications where understanding the flow or distribution within a network is crucial. These curvature-based approaches have opened new avenues in network analysis, offering a geometric perspective to complement traditional topological and statistical methods.

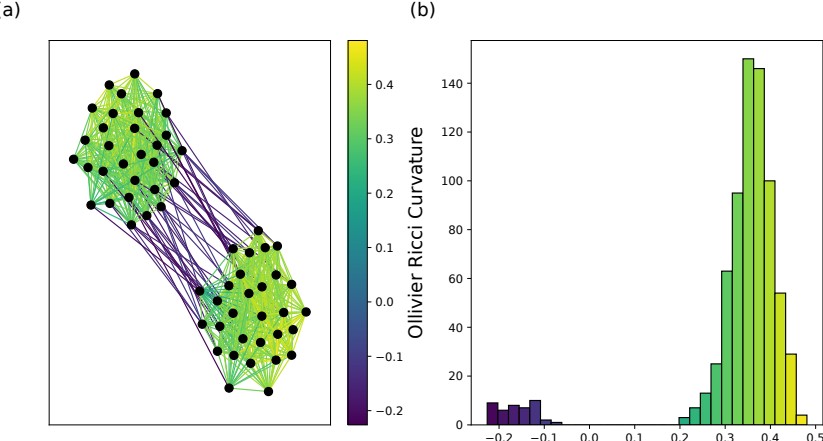

Figure 1: (a) A toy simulated network with two communities, with edges colored by ORC. (b) The histogram of ORC, suggesting its potential in community detection.

Although discrete curvature has found various applications in network analysis, such as understanding internet topology (Ni et al., 2015), differentiating cancer networks (Sandhu et al., 2015), and addressing oversquashing and oversmoothing problems in graph neural networks (Nguyen et al., 2023), its specific use in community detection remains relatively underexplored. Figure 1 shows a toy network with two communities. In this network, the edges connecting different communities tend to have lower ORC values than those within a community. This observation shows the potential of network curvatures in community detection. In relation to this observation, Sia et al. (2019) proposed to iteratively remove the edge with the smallest ORC and recalculating all edge ORCs until the network becomes disconnected, with each connected component identified as a community. Similarly, Fesser et al. (2023) proposed another iterative algorithm to remove edges with augmented FRC above a threshold, until no edge curvature exceeds that threshold.

However, these approaches have several major drawbacks. First, the computational cost of calculating ORC and augmented FRC of a single edge is high, scaled as $O(n^3)$ for ORC and $O(n^2)$ for augmented FRC, where $n$ represents the number of nodes. This makes it prohibitively expensive for large scale such as the DBLP co-authorship network ($n = 317,080$), the Amazon product co-purchasing network ($n = 334,863$), and the YouTube social network ($n = 1,134,890$, Yang & Leskovec (2012)). Second, the iterative nature of the algorithm introduces significant extra time inefficiencies. In the worst-case scenario, up to $O(m)$ iterations might be required. Third, the methods, despite their innovative approach, can sometimes be too restrictive and may underperform compared to popular methods such as the Leiden algorithm.

To effectively tackle the challenges in community detection within large-scale networks, our study introduces a novel curvature measure, the Lower Ricci Curvature (LRC). LRC is specifically designed for efficient computation, with a linear computational complexity of $O(n)$. This significantly reduces the computational burden compared to traditional curvature measures, making it highly suitable for large networks. In addition to its computational efficiency, we provide some theoretical analysis of LRC, particularly its connection to the Cheeger constant, a well-established concept in graph theory (Mohar, 1989), which helps in understanding how LRC relates to the division of a network into communities.

Building on the theoretical foundation of LRC, we have developed a preprocessing algorithm that utilizes LRC to improve existing community detection methods. This algorithm is designed to be suitable for a wide range of applications, due to its adaptability to different network structures and sizes and its compatibility with various community detection methods. We applied it to networks of diverse sizes, including both small-scale networks (NCAA football league network) and large-scale networks with mixed membership (the DBLP coauthorship network, the Amazon product co-purchasing network, and the YouTube social network). The results from these studies consistently demonstrate that our preprocessing step, based on the Lower Ricci Curvature, not only enhances the efficiency but also improves the accuracy of various established community detection methods.

Proofs, simulation results, and additional experimental details are provided in the Appendix.

## 2 Background

### 2.1 Community detection

Community detection in network analysis is essential for unraveling the intricate structures of networks. Communities are typically defined as subgroups of nodes with denser internal connections compared to their external connections (Radicchi et al., 2004). Understanding these communities is vital to reveal the underlying structural characteristics of networks and to classify nodes based on their interrelations (Fortunato & Hric, 2016).

While the Introduction briefly mentions various community detection algorithms, this subsection aims to delve deeper into their specific functionalities and contributions. The Girvan-Newman algorithm leverages edge betweenness centrality and hierarchical clustering to identify community structures (Newman & Girvan, 2004). The Leiden algorithm, evolving from the Louvain method, focuses on optimizing modularity, thereby enhancing the resolution and accuracy of detected communities (Blondel et al., 2008; Traag et al., 2019). Other notable methods include Label Propagation, which relies on the diffusion of information (Raghavan

et al., 2007), and Walktrap, which uses random walks to discern community structures (Pons & Latapy, 2005).

Additionally, algorithms such as the Angel (Rossetti, 2020), ego-based community detection (Ego, Leskovec & Mcauley (2012)), K-clique (Palla et al., 2005), Speaker-Listener Label Propagation Algorithm (SLPA, Xie et al. (2011)) contribute diverse perspectives and techniques to community detection. Each of these methods brings unique strengths to the analysis of network structures, addressing different aspects and challenges in identifying community patterns.

To evaluate the performance of these algorithms, criteria such as the Adjusted Rand Index (ARI) and the Adjusted Mutual Information (AMI) are commonly used. ARI evaluates the agreement in node pair assignments between clustering results, offering a quantitative assessment of similarity (Rand, 1971). AMI measures the similarity between different community detection results on the same dataset, providing insights into the amount of shared information (Vinh et al., 2009). In our study, we utilize these criteria to gauge the improvements in community detection algorithms' performance when incorporating our newly proposed LRC-based preprocessing step.

## 2.2 Discrete curvatures

Curvature, a fundamental concept in mathematics, describes how a curve deviates from a straight line or a surface from being flat (Boothby, 1986). In Riemannian geometry, curvatures such as Ricci curvature, provide insights into the unique geometry properties of different spaces, including volume changes along geodesics (Do Carmo & Flaherty Francis, 1992). Historically, generalizing this concept to discrete objects, such as networks, presented a significant challenge. A pivotal moment in this endeavor came with the work of Forman (Forman, 2003), who innovatively adapted curvature concepts to the discrete realm. This milestone opened the door for further exploration and application of curvatures in discrete spaces, including networks.

For presentational simplicity, in this paper, we focus on an unweighted graph $G = (V, E)$, where $V$ is a set of nodes and $E$ is a set of edges, but the framework can be generalized to a weighted graph in a straightforward manner. Let $(ij)$ be an edge connecting node $i$ and node $j$, we denote the degree of $i$, i.e., the number of edges of node $i$, by $n_i$, the number of shared neighbors of $i, j$, i.e., the number of triangles based on $(ij)$, by $n_{ij}$. Under this framework, the Forman Ricci Curvature, or FRC, is defined as follows:

**Definition 1 (Forman Ricci Curvature (FRC))** *The FRC of edge $(ij)$ is defined as*

$$\mathrm{FRC}(ij) = 4 - n_i - n_j + 3n_{ij}.$$

FRC evaluates the "structural support" of an edge by presenting higher values to edges with more shared neighbors and lower-degree endpoints, since they are more "integral" to the network's structure. The computational cost for calculating FRCs for a single edge is O($n$). Following Forman's groundbreaking work (Forman, 2003), there has been a surge of studies exploring and applying FRC, to various network structures. For example, Sreejith et al. (2016a) extends FRC from undirected to directed networks, and Sreejith et al. (2016b) extends FRC to complex networks and explains its connection to continuous Ricci curvature.

However, its unbounded and scale-dependent nature, as well as its skewness toward negative values in various networks pose interpretational challenges (Sreejith et al., 2016b). To address these issues, an improved version known as the Balanced Forman Curvature (BFC, Topping et al. (2021)) was proposed:

**Definition 2 (Balanced Forman Curvature (BFC))** *The BFC of edge $(ij)$ is defined as*

$$\mathrm{BFC}(ij) = \frac{2}{n_i} + \frac{2}{n_j} - 2 + 2\frac{n_{ij}}{\max(n_i, n_j)} + \frac{n_{ij}}{\min(n_i, n_j)} + \frac{s_{i,j} + s_{j,i}}{\gamma_{\max} \max(n_i, n_j)},$$

*where $s_{i,j}$ is the number of neighbors of node $i$ forming a 4-cycle based at the edge $(ij)$ without diagonals inside, $\gamma_{\max}$ is the maximal number of 4-cycles based at edge $(ij)$ traversing a common node (see Topping et al. (2021) for more details).*

BFC rescales the contribution of shared neighbors and node degrees, making it bounded within the range $[-2, 2]$. This modification ensures that the curvature captures essential topological features, such as bottlenecks, more consistently. With this feature, BFC has been applied to identify the bottleneck structure in graphs, particularly for addressing the over-squashing phenomenon in graph neural networks. However, its computational complexity of $O(n^2)$, limits its application to large-scale networks, mainly due to computationally intensive terms $s_{i,j}$ and $\gamma_{\max}$, which involves counting the number of squares based at nodes $i$ and $j$ under certain constraints (Topping et al., 2021).

Moreover, Ollivier made significant contributions by defining the Ricci curvature for networks through optimal transport and differential equations (Ollivier, 2007):

**Definition 3 (Ollivier-Ricci Curvature (ORC))** *The ORC of edge $(ij)$ is defined as*

$$\text{ORC}(ij) = 1 - \frac{W_1(m_i, m_j)}{d(i, j)},$$

*where $W_1$ is Wasserstein-1 distance, $m_i$ is a local probability measures at node $i$, defined as*

$$m_i(k) = \begin{cases} \frac{1}{n_i}, & (ik) \in E \\ 0, & (ik) \notin E \end{cases} \tag{1}$$

*and $d(i, j)$ is the length of a shortest path from node $i$ to node $j$, also known as the graph distance.*

ORC quantifies the change in probability distributions across the neighborhoods of two nodes connected by an edge. A larger ORC value indicates stronger cohesion between the two neighborhoods, while a smaller or negative value suggests a bridge-like connection. This curvature has sparked a wide array of follow-up research, further enriching the field of network analysis with these novel curvature-based insights. For example, Lin & Yau (2010); Lin et al. (2011); Erbar & Maas (2012); Bauer et al. (2013) have provided deep mathematical insights into the properties and implications of ORC in the context of graph theory and network geometry. Trillos & Weber (2023) showed its connection to continuous Ricci curvature using random geometric graph sampled from a manifold. Notably, Sia et al. (2019) utilized ORC for community detection, iteratively removing the edge with the smallest ORC. However, its computational cost ($O(n^3)$) poses significant challenges, especially for iterative algorithms.

## 2.3 Stochastic Block Model (SBM)

To illustrate network curvatures in a simplified context, we consider the Stochastic Block Model (SBM) in this manuscript, a basic yet versatile model used in network analysis (Holland et al., 1983). SBM is renowned for its ability to mimic community structures within networks. In this model, nodes are partitioned into $K$ distinct communities, and connections between nodes are probabilistically determined based on their community memberships.

Each node $i$ is assigned a community label $z_i \in \{1, \cdots, K\}$, indicating its community membership. The block matrix $B$, a $K \times K$ symmetric matrix, is a critical component of the SBM, dictating the probability of edge formation between nodes from communities. Specifically, $B_{kl}$ represents the probability of an edge existing between nodes from community $k$ and community $l$. In an SBM, the probability of an edge existing between any two nodes $i$ and $j$ follows a Bernoulli distribution, and is independent of other edges, as reflected in the adjacency matrix $A \in \mathbb{R}^{n \times}$ :

$$\mathbb{P}(A_{ij} = 1) = B_{z_i z_j}$$

While a basic SBM might appear too simplistic for complex real-world data, its extensions, such as hierarchical SBM and mixed membership SBM, offer more nuanced representations. These models cater to scenarios involving hierarchical community structures (Peixoto, 2014) or nodes with memberships in multiple groups (Airoldi et al., 2008).

## 3 Lower Ricci Curvature (LRC)

In this section, we introduce a novel discrete curvature, the Lower Ricci Curvature (LRC), notably for its high performance in community detection and low computational complexity of $O(n)$. We delve into the intuition behind defining LRC, and how these factors contribute to its computational efficiency and efficacy in community detection.

The LRC of edge $(ij)$ is defined as:

$$\text{LRC}(ij) = \frac{2}{n_i} + \frac{2}{n_j} - 2 + 2\frac{n_{ij}}{\max(n_i, n_j)} + \frac{n_{ij}}{\min(n_i, n_j)}.$$

Several key observations about LRC can be made. Firstly, the computation of $\text{LRC}(ij)$ requires $O(n)$, similar to FRC. Secondly, LRC is always within the range of $[-2, 2]$, aligning with the bounds of BFC. Third, LRC is consistently less than or equal to BFC, with the difference, denoted as $\Delta(ij)$, defined as

$$\Delta(ij) := \frac{s_{i,j} + s_{j,i}}{\gamma_{\max} \max(n_i, n_j)} = \text{BFC}(ij) - \text{LRC}(ij) \geq 0.$$

In fact, BFC is further upper bounded by ORC (Topping et al., 2021), leading to the following proposition, which underlines why we term it *lower* Ricci curvature.

**Proposition 1** *For any edge $(ij)$,*

$$\text{LRC}(ij) \leq \text{BFC}(ij) \leq \text{ORC}(ij).$$

This proposition motivates our first rationale for defining LRC. The computational bottleneck of BFC is the term $\Delta$, which requires $O(n^2)$ time. However, this leads to two pertinent questions. First, is LRC effective in differentiating edges within and between communities? Second, does the term $\Delta$ significantly contribute to community detection, or is there a notable difference in $\Delta$ for edges within the same community versus those between different communities?

To further investigate these questions, we utilize an SBM-generated network as a toy example. Figure 2 presents a network with $n = 60$ nodes, divided evenly into two communities. Edges within communities have a higher probability of 0.8, while edges between communities are set at a lower probability of 0.05. The edges are color-coded based on their LRCs: higher LRCs are marked in yellow, while lower LRCs are marked in purple. This visual representation helps highlight that edges bridging different communities tend to have smaller LRC values compared to those within the same community. This example shows that LRC achieves computational efficiency by omitting the computationally expensive term $\Delta$ without sacrificing its ability to detect community structures.

The direct link between LRC and ORC is less straightforward, which guides the second intuition behind the definition of LRC. The primary computational challenge in calculating ORC lies in the Wasserstein-1 distance, also known as the earth moving distance (Villani et al., 2009). A natural approach is to approximate this distance or bound it from below, above, or both. To effectively bound ORC, it is necessary to bound the Wasserstein distance, which involves complex calculations of the total cost of certain candidate transports (see Jost & Liu (2014) for more details). The bounds are established as follows:

$$\frac{2}{n_i} + \frac{2}{n_j} - 2 + 2\frac{n_{ij}}{\max(n_i, n_j)} + \frac{n_{ij}}{\min(n_i, n_j)} \leq \text{ORC}(ij) \leq \frac{n_{ij}}{\max\{n_i, n_j\}}.$$

Notably, the lower bound is precisely the LRC, which connects back to Proposition 1. While it is technically feasible to use the upper bound to define a curvature, it has some drawback. First, the upper bound is always positive, which conflicts with the intuition that Ricci curvature can be positive (sphere), negative (hyperbolic space), or zero (hyperplane), reflecting different geometric properties. Second, some theoretical support, as shown in the Corollary below, does not apply to this upper bound. Third, it has some limitations on distinguishing between different network structures. For example, if there is only one edge connecting

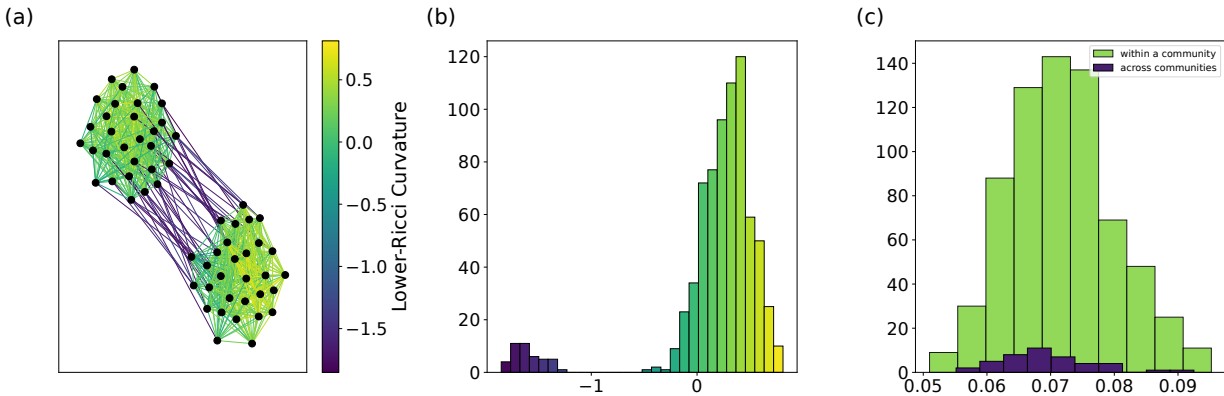

Figure 2: (a) A SBM-generated network with $K = 2$, $B_{kk} = 0.8$, $B_{kl} = 0.05$ for $k \neq l$, with edges colored by LRC. (b) The histogram of LRC, suggesting its potential in community detection. (c) The histogram of $\Delta$ for within and across community edges, indicating that $\Delta$ may not significantly contribute to community detection.

two fully connected communities, then $\frac{n_{ij}}{\max\{n_i, n_j\}} = 0$, which is same as any edge in a straight-line graph. Hence, we focus on the lower bound, i.e., LRC.

As a direct corollary of these inequalities, the following corollary establishes a link between the bound of LRC and the diameter of the network, defined as $\mathrm{diam}(G) = \sup_{i,j \in V} d(i,j)$, where $d$ represents the graph distance. This is also related to the Cheeger constant (Chung, 1997), a measure indicative of the presence of a community structure in the network.

**Corollary 1** *If there exists $\alpha > 0$ such that $\mathrm{LRC}(ij) \geq \alpha$ for any $(ij) \in E$, then*

1. $\mathrm{diam}(G) \leq \frac{2}{\alpha}$.

2. $\frac{\lambda_1}{2} \geq h_G \geq \frac{\alpha}{2}$, *where $\lambda_1$ is first non-zero eigenvalue of the normalized graph Laplacian, also known as the spectral gap, and $h_G$ is the Cheeger constant.*

The interpretation of this corollary is that a larger value of $\alpha$, suggests a graph structure more akin to a fully connected graph, hence a smaller diameter. Similarly, a larger $\alpha$ correlates with a higher Cheeger constant, indicating a more interconnected network with less pronounced separability into distinct community structures.

We conclude this section with Table 1 to compare four curvatures: FRC, BFC, ORC, and LRC. The computational complexity is the cost of calculating the curvatures for each edge. Scale-free means the curvature is independent of the network size characterized by $n$ and $m$. Scale-free properties are particularly important in the network analysis, as they ensure the applicability and consistency of curvature measures across networks of different sizes. This quality is preferable as it allows for meaningful comparisons (Sandhu et al., 2015) and generalizations across various network structures, from small-scale to large-scale networks, without being biased by their size. FRC is not scale-free because its values can grow unbounded in large networks due to its definition.

Among these curvatures, LRC stands out for its linear computational complexity and scale-free property, making it a versatile and efficient choice for network analysis. This blend of computational efficiency and scale-free nature makes LRC an ideal candidate for analyzing networks in various contexts. While this paper primarily focuses on the practical applications of LRC, some discussion on its theoretical aspects, including concentration of $n_{ij}$ and phase transitions in SBM, can be found in Appendix B and Appendix C.1

Table 1: Comparison of four curvatures.

| Curvature | Computational Complexity | Scale-Free |
|-----------|--------------------------|------------|
| FRC | $\mathbf{O(n)}$ | No |
| BFC | $O(n^2)$ | **Yes** |
| ORC | $O(n^3)$ | **Yes** |
| LRC | $\mathbf{O(n)}$ | **Yes** |

## 4 LRC-based preprocessing

As observed in Figure 2(b), the presence of community structures in networks often results in a bimodal distribution of LRC values. This typically manifests itself as two distinct modes in the histogram of LRCs: a smaller mode corresponding to across-community edges and a larger mode representing within-community edges. This observation underpins our proposed preprocessing step for community detection: removing edges with small LRC values below a specific threshold. This approach aims to retain more within-community edges, thereby making the community structure more pronounced. The threshold is determined by fitting a Gaussian mixture model (GMM, Reynolds et al. (2009)) to the histogram of LRCs, as outlined in the following algorithm.

---
**Algorithm 1:** LRC-based preprocessing algorithm for community detection
---
**Input:** Raw network data: $G = (V, E)$
**Output:** Preprocessed network data $G' = (V, E')$
**1** Calculate the LRC for all edges;
**2** Fit a Gaussian mixture model with two mixing component to LRCs, obtaining the estimate
$\hat{p}(x) = \pi_1 N(x; \mu_1, \sigma_1^2) + \pi_2 N(x; \mu_2, \sigma_2^2)$, where $\mu_1 < \mu_2$;
**3** Find the local minimum $\beta := \inf_{\mu_1 < x < \mu_2} \hat{p}(x)$;
**4** Remove all edges with LRCs smaller than $\beta$: $E' := \{(ij) \in E : \mathrm{LRC}(ij) \geq \beta\}$
---

The workflow of our proposed method is illustrated in Figure 3, based on the network example previously discussed.

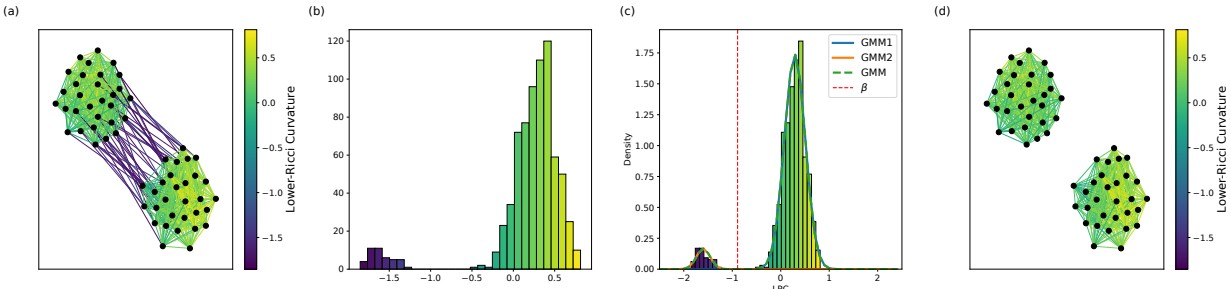

Figure 3: (a) A SBM-generated network with $K = 2$, $B_{kk} = 0.8$, $B_{kl} = 0.05$ for $k \neq l$, with edges colored by LRC. (b) The histogram of LRC, with each bar colored by LRC. (c) The threshold $\beta$ (the dotted vertical line) estimated by GMM. GMM1 is the mixing component with a large mean $\mu_2$, GMM2 is the mixing component with a smaller mean $\mu_1$. (d) The processed network, exhibiting a more discernible community structure.

This toy example illustrates how our preprocessing step is expected to enhance the performance of existing community detection methods by clarifying the underlying community structures.

Following the description of the underlying community structures, it is crucial to highlight the efficiency and scalability of the LRC-based preprocessing method. The calculation of LRC itself requires O($n$) time, and the subsequent edge removal step is a one-time, non-iterative process. This is in stark contrast to competitor methods that rely on iterative processes (Jost & Liu, 2014; Sia et al., 2019), which can significantly increase computational time, especially for large networks that are increasingly common in various domains.

Crucially, this increase in efficiency does not compromise the accuracy of community detection. In the Appendix, we provide the result of simulation study using SBM-generated networks to evaluate the effectiveness of LRC on community detection. In the following section, we present empirical evidence showing how our LRC-based preprocessing not only maintains, but often enhances the effectiveness of popular community detection algorithms, even in complex network scenarios. This demonstrates the dual benefit of our approach: it streamlines computation while enriching the depth of network analysis.

While LRC preprocessing may seem similar to a denoising method (Yu et al., 2021), it is not intended to treat low-curvature edges (e.g., bridges) as noise. Instead, these edges are considered meaningful structural features whose removal can enhance community detection. See the Appendix D for a detailed discussion on the distinction from denoising approaches.

## 5  Application

In this section, we evaluated the impact of our proposed LRC preprocessing method on the performance of various community detection algorithms using four real-world datasets with known community structures. We begin our analysis with a smaller network, the NCAA Football League Network, to demonstrate the impact of our preprocessing method in a more controlled setting. To this end, we compared the Adjusted Rand Index (ARI) and Adjusted Mutual Information (AMI) scores before and after applying our preprocessing technique with LRC, utilizing four representative community detection models: Label Propagation, Leiden, Girvan-Newman, and Walktrap. These models were chosen for their effectiveness and widespread use in community detection, as noted in the existing literature (Fortunato & Hric, 2016). In addition to the LRC-based preprocessing method, we also applied the preprocessing algorithm using BFC and FRC to further demonstrate the performance of curvature-based preprocessing in a broader framework. We did not include ORC in these experiments, due to its high computational complexity.

After evaluating the NCAA Football League Network, we extend our analysis to three larger-scale networks. These networks pose additional challenges, particularly in terms of computational efficiency. Moreover, they often exhibit mixed membership, where nodes can belong to multiple communities, diverging from the unique community structures seen in smaller and simpler networks like the NCAA dataset.

Given these differences, we shift our focus to algorithms better suited for these conditions. For larger networks, we use Angel, Ego, K-clique, and Speaker-Listener Label Propagation Algorithm (SLPA). These replacements are due to the suitability of these algorithms in handling large-scale networks and their capability to address mixed membership scenarios. In addition, since BFC requires a significant amount of computational time with a large memory size for calculating curvatures in large networks, we applied only FRC and LRC preprocessing on the three following networks.

Furthermore, since the ARI and AMI scores are less effective for evaluating community detection in networks with mixed memberships, we utilize the F1 score to evaluate community detection results. This is a well-established metric in such scenarios because it combines the precision and recall of the detected communities to provide a balanced measure of a method's accuracy and is particularly useful in networks where a node can be part of multiple communities (Hollocou et al., 2018).

### 5.1  NCAA Football League Network

The NCAA Football League Network is a graph representing the NCAA Division I football game schedule in the year 2000 (Girvan & Newman, 2002). There are a total of 115 nodes, each representing college football teams, and 613 edges, corresponding to the regular season games played between these teams. Each node in the network is assigned to one of the 12 ground-truth conference groups. Since teams tend to play more

frequently within their own conference groups, each conference in the network clearly exhibits a community structure. Table 2 below illustrates the performance improvement of various community detection algorithms through the application of our preprocessing method.

Table 2: Community detection algorithms evaluation for NCAA Football League network

| Scores | Algorithms | | | |
| --- | --- | --- | --- | --- |
| | Label Propagation | Leiden | Girvan-Newman | Walktrap |
| ARI: before | 0.75 | 0.81 | 0.84 | 0.82 |
| ARI: after BFC | 0.90 | 0.90 | 0.84 | 0.90 |
| ARI: after FRC | 0.90 | 0.90 | 0.83 | 0.90 |
| ARI: after LRC | 0.90 | 0.90 | 0.84 | 0.90 |
| AMI: before | 0.83 | 0.86 | 0.87 | 0.86 |
| AMI: after BFC | 0.90 | 0.90 | 0.88 | 0.90 |
| AMI: after FRC | 0.90 | 0.90 | 0.87 | 0.90 |
| AMI: after LRC | 0.90 | 0.90 | 0.88 | 0.90 |
| Complexity | $O(m)$ | $O(nm)$ | $O(nm^2)$ | $O(n)$ |

The results clearly show an improvement in both ARI and AMI scores after applying our preprocessing method based on network curvatures, across all four community detection algorithms. Comparing the three curvatures, LRC-based processing achieved similar AMI and ARI compared to BFC and FRC-based preprocessing. Notably, the Label Propagation algorithm, which initially had the lowest ARI (0.75) and AMI (0.83), significantly improved to both 0.90, respectively, after preprocessing. This enhancement elevates it to one of the top-performing algorithms in this context. In fact, after preprocessing, all algorithms exhibit very similar scores, suggesting that our preprocessing method regardless of the type of curvature simplifies the community detection problem by making the community structures more distinct and evident. This homogenization of performance implies that, with effective preprocessing, the choice of community detection algorithm becomes less critical, as the clarified network structure facilitates more accurate community detection across different methods.

Importantly, the inclusion of computational complexity in our analysis (as shown in the last row of the table) provides further insight into the selection of an optimal algorithm. Algorithms such as Label Propagation and Walktrap, with their lower computational complexities of $O(m)$ and $O(n)$ respectively, become attractive options. This highlights another significant advantage of our preprocessing method – it not only improves the accuracy of community detection, but also enhances overall efficiency by enabling the use of faster algorithms without compromising on performance.

## 5.2 DBLP Collaboration Network

The DBLP Collaboration Network is a graph that shows coauthorship relationships in the computer science bibliography between multiple researchers (Yang & Leskovec, 2012). The network comprises 317,080 nodes representing researchers and 1,049,866 edges indicating a collaborative paper. The results, as shown in the table below, demonstrate that the LRC preprocessing method aids in detecting community structures in more complex networks.

For methods in Table 3, the traditional Big O notation for computational complexity are not consistently available in the literature. Instead, we focus on the actual runtime of each algorithm, providing a more practical measure of efficiency in real-world applications. The 'Time: before' represents the runtime (in seconds) of each community detection algorithm when applied directly to the raw network data, while 'Time: after' encompasses the total time, which includes calculating the curvature, identifying the threshold, removing edges based on this threshold, and rerunning the same community detection algorithm on the processed network. This approach ensures a fair comparison, as it accounts for all steps involved in our preprocessing method. It is important to note that we do not claim our method to be faster than the

Table 3: Community detection algorithms evaluation for DBLP network

| | Algorithms | | | |
|---|---|---|---|---|
| Scores | Angel | Ego | K-clique | SLPA |
| F1: before | 0.284 | 0.317 | 0.276 | 0.211 |
| F1: after FRC | 0.385 | 0.311 | 0.383 | 0.334 |
| F1: after LRC | **0.452** | 0.386 | 0.420 | 0.371 |
| Time: before | 260.17 | 831.87 | 40.50 | 1024.12 |
| Time: after FRC | 168.03 | 122.03 | 85.03 | 549.03 |
| Time: after LRC | **180.39** | 184.88 | 111.60 | 2349.36 |

approach without preprocessing. Instead, the table shows that the preprocessing step has a minimal impact on the total computational time, given its linear complexity.

Table 3 reveals a notable improvement in the F1 scores for each algorithm after integrating our preprocessing method to DBLP collaboration network. Although preprocessing with FRC tends to increase the accuracy of community detection, preprocessing with LRC shows more significant improvement overall. Furthermore, this improvement in performance does not come at the cost of reduced efficiency; in fact, the Angel algorithm demonstrates increased processing speed post-preprocessing, even with these additional preprocessing steps, highlighting the efficiency of our method in complex network environments.

### 5.3 Amazon Product Co-purchasing Network

This network represents the co-purchasing patterns of products on Amazon (Yang & Leskovec, 2012). The nodes symbolize products, and the edges indicate co-purchases by Amazon customers. The network includes 334,863 nodes and 925,872 edges. As with the previous datasets, the application of the LRC preprocessing method significantly enhanced the results of various community detection algorithms, as illustrated in the table below.

Table 4: Community detection algorithms evaluation for the Amazon network

| | Algorithms | | | |
|---|---|---|---|---|
| Scores | Angel | Ego | K-clique | SLPA |
| F1: before | 0.368 | 0.371 | 0.387 | 0.345 |
| F1: after FRC | 0.375 | 0.374 | 0.394 | 0.351 |
| F1: after LRC | 0.463 | 0.444 | **0.482** | 0.483 |
| Time: before | 159.52 | 1000.85 | 42.40 | 2380.61 |
| Time: after FRC | 160.59 | 112.59 | 67.59 | 810.59 |
| Time: after LRC | 139.61 | 629.56 | **85.73** | 3911.19 |

In line with the results of the DBLP collaboration network, our method utilizing LRC significantly improved the performance of each community detection algorithm.

### 5.4 YouTube Social Network

This dataset represents the social network on YouTube (Mislove et al., 2007), where nodes symbolize users, and edges indicate subscription between YouTube users. The network includes the total number of 1,134,890 nodes and 2,987,624 edges. The results show that the implementation of the LRC preprocessing method greatly improved the results of multiple community detection algorithms in line with prior datasets.

Table 5 showcases the effectiveness of our preprocessing method using LRC in the YouTube social network. While the performance boost is apparent across all algorithms, the Ego and SLPA algorithms stand out for

Table 5: Community detection algorithms evaluation for the YouTube network

|  | Algorithms | | | |
| Scores | Angel | Ego | K-clique | SLPA |
|---|---|---|---|---|
| F1: before | 0.063 | 0.223 | 0.066 | 0.093 |
| F1: after FRC | 0.086 | 0.238 | 0.091 | 0.124 |
| F1: after LRC | 0.282 | **0.445** | 0.216 | 0.429 |
| Time: before | 972.74 | 143.09 | 9029.98 | 117.23 |
| Time: after FRC | 392.30 | 131.30 | 1895.71 | 1974.30 |
| Time: after LRC | 67.84 | **129.63** | 131.75 | 218.29 |

their marked improvements in F1 scores. This result diverges slightly from other large networks, as K-clique is not the fastest method here. Nevertheless, our method consistently enhances the overall performance of community detection, particularly benefiting faster analysis methods.

In conclusion, curvature-based preprocessing algorithm consistently improves community detection results on real-world datasets. In particular, our LRC preprocessing method demonstrates its strength on large complex networks. Across all three large networks analyzed – DBLP, Amazon, and YouTube – our LRC preprocessing method consistently enables at least one community detection algorithm to achieve the best or near-best performance scores, while maintaining impressive efficiency with runtimes under 200 seconds. For instance, the Angel algorithm for the DBLP network, K-clique for the Amazon network, and Ego for the YouTube network each emerged as top performers in their respective datasets. This is particularly noteworthy given the substantial size of these networks. Such results underscore the exceptional effectiveness and efficiency of our LRC preprocessing approach in handling complex, large-scale network data, making it a highly valuable tool in the field of network analysis. Moreover, the LRC preprocessing could be applied to any community detection algorithm. In particular, fast community detection methods, such as Wang et al. (2020), can be combined with LRC preprocessing for efficient and effective community detections.

## 6   Discussion and future work

In this work, we have focused on network curvature and its applications in community detection. Our key contribution is the proposal of the Lower Ricci Curvature (LRC), a scalable and scale-free discrete curvature designed specifically for network analysis. Alongside this, we have developed an LRC-based preprocessing method that has shown potential in enhancing the performance of established community detection methods. This assertion is backed by multiple real-world applications, including analyses of large-scale networks such as Amazon, DBLP, and YouTube. Moreover, the LRC framework is adaptable and can be straightforwardly extended to weighted networks. Looking forward, several promising directions for extending this research are evident.

**Extension to directed graphs**: Extending LRC to directed graphs opens up numerous possibilities for analysis in various fields. Directed graphs are crucial in representing asymmetric relationships, such as citation networks in academia, where the directionality of citations plays a significant role (Newman, 2001), or in web link structures where the direction of links implies a flow of information (Kleinberg et al., 1999). Adapting LRC to account for the directionality in such networks can provide more nuanced insights into their structural and community dynamics.

**Application to hypergraphs**: Hypergraphs, which involve higher-order interactions beyond pairwise connections, present an exciting frontier. For instance, in collaborative environments like multi-author scientific publications (Taramasco et al., 2010) or gene co-expression  (Tran, 2012), interactions are inherently multi-dimensional. Extending LRC to hypergraphs could yield a deeper understanding of these complex relational structures and the underlying community formations.

**Deeper theoretical investigation of LRC**: A systematic study of network curvature's theoretical properties under different models, including but not limited to SBM, is an important direction for future work. Based on the theoretical insights, exploring alternative curvature definitions and analyzing their effectiveness under different model settings, could further improve the performance curvature-based methods in community detection.

## 7 Acknowledgments

YJP was supported by NIH grant T32 CA106209; DL was supported by NIH grants P30 ES010126, R01 HL149683, R01 HL173044, R01 LM014407, R56 LM013784, UM1 TR004406.

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

## Appendix

## A    Proof of Proposition 1 and Corollary 1

$LRC \le BFC$ directly follows from definition, as $BFC - LRC = \Delta \ge 0$. The inequality $BFC \le ORC$ is derived from Theorem 2 in Topping et al. (2021). Corollary 1 is a direct consequence of Proposition 1 combined with Corollary 3 and Proposition 5 from Topping et al. (2021).

## B    Concentration for $n_{ij}$ under SBM

In this section, we aim to compute the probability of all within-community $n_{ij}$ values are greater than all across-community $n_{ij}$ values in a simple case of an SBM with two communities, each of size $n$:

$$\min_{(i,j)\in\text{within}} n_{ij}^{\text{within}} > \max_{(i,j)\in\text{across}} n_{ij}^{\text{across}}.$$

We first calculate the expected values of $n_{ij}$ separately for within-community and across-community edges. For nodes $i, j$ in the same community:

$$\mathbb{E}[n_{ij}^{\text{within}}] = (n-2)p_1^2 + np_2^2,$$

For nodes $i, j$ in different communities:

$$\mathbb{E}[n_{ij}^{\text{across}}] = 2(n-1)p_1p_2,$$

Using Chernoff bounds,

$$\Pr\left(n_{ij}^{\text{within}} < (1-\epsilon)\mathbb{E}[n_{ij}^{\text{within}}]\right) \leq \exp\left(-\frac{\epsilon^2 \mathbb{E}[n_{ij}^{\text{within}}]}{2}\right).$$

Applying the union bound, we have

$$\Pr\left(\forall(i,j) \in \text{within}, n_{ij}^{\text{within}} \geq (1-\epsilon)\mathbb{E}[n_{ij}^{\text{within}}]\right) \geq 1 - 4n^2 \cdot \exp\left(-\frac{\epsilon^2 \mathbb{E}[n_{ij}^{\text{within}}]}{2}\right).$$

Similarly, for across-community edges, using Chernoff bounds,

$$\Pr\left(n_{ij}^{\text{across}} > (1+\epsilon)\mathbb{E}[n_{ij}^{\text{across}}]\right) \leq \exp\left(-\frac{\epsilon^2 \mathbb{E}[n_{ij}^{\text{across}}]}{3}\right).$$

Using the union bound,

$$\Pr\left(\forall(i,j) \in \text{across}, n_{ij}^{\text{across}} \leq (1+\epsilon)\mathbb{E}[n_{ij}^{\text{across}}]\right) \geq 1 - 4n^2 \cdot \exp\left(-\frac{\epsilon^2 \mathbb{E}[n_{ij}^{\text{across}}]}{3}\right).$$

To ensure perfect recovery, we set $\epsilon$ to ensure non-overlap between within-community and across-community bounds:

$$(1-\epsilon)\mathbb{E}[n_{ij}^{\text{within}}] > (1+\epsilon)\mathbb{E}[n_{ij}^{\text{across}}].$$

Substitute the expectations:

$$(1-\epsilon)((n-2)p_1^2 + np_2^2) > (1+\epsilon)(2(n-1)p_1p_2).$$

Solving for $\epsilon$:

$$\epsilon < \frac{(n-2)p_1^2 + np_2^2 - 2(n-1)p_1p_2}{(n-2)p_1^2 + np_2^2 + 2(n-1)p_1p_2}.$$

To create a safety margin, we set:

$$\epsilon = \frac{1}{2} \cdot \frac{(n-2)p_1^2 + np_2^2 - 2(n-1)p_1p_2}{(n-2)p_1^2 + np_2^2 + 2(n-1)p_1p_2}.$$

Substitute this $\epsilon$ into the union bounds. For within-community edges:

$$\Pr\left(\exists(i,j) \in \text{within}, n_{ij}^{\text{within}} < (1-\epsilon)\mathbb{E}[n_{ij}^{\text{within}}]\right) \leq 4n^2 \cdot \exp\left(-\frac{\epsilon^2((n-2)p_1^2 + np_2^2)}{2}\right).$$

For across-community edges:

$$\Pr\left(\exists(i,j) \in \text{across}, n_{ij}^{\text{across}} > (1+\epsilon)\mathbb{E}[n_{ij}^{\text{across}}]\right) \leq 4n^2 \cdot \exp\left(-\frac{\epsilon^2(2(n-1)p_1p_2)}{3}\right).$$

The total failure probability is:

$$\Pr(\text{Failure}) \leq 4n^2 \cdot \exp\left(-\frac{\epsilon^2((n-2)p_1^2 + np_2^2)}{2}\right) + 4n^2 \cdot \exp\left(-\frac{\epsilon^2(2(n-1)p_1p_2)}{3}\right).$$

Substitute $\epsilon = \frac{1}{2} \cdot \frac{(n-2)p_1^2 + np_2^2 - 2(n-1)p_1p_2}{(n-2)p_1^2 + np_2^2 + 2(n-1)p_1p_2}$ into both terms:

$$\Pr(\text{Failure}) \leq 4n^2 \cdot \exp\left(-\frac{\left((n-2)p_1^2 + np_2^2 - 2(n-1)p_1p_2\right)^2}{8\left((n-2)p_1^2 + np_2^2 + 2(n-1)p_1p_2\right)^2} \cdot \left((n-2)p_1^2 + np_2^2\right)\right)$$

$$+ 4n^2 \cdot \exp\left(-\frac{\left((n-2)p_1^2 + np_2^2 - 2(n-1)p_1p_2\right)^2}{12\left((n-2)p_1^2 + np_2^2 + 2(n-1)p_1p_2\right)^2} \cdot (2(n-1)p_1p_2)\right).$$

As $n \to \infty$, both terms decay exponentially, ensuring:

$$\Pr(\text{Failure}) \to 0.$$

However, these calculations are specific to $n_{ij}$ and do not fully extend to LRC, which involves additional dependencies on $n_i$ and $n_j$. Rigorous analysis of LRC under SBM requires further investigation, as it introduces additional complexity. A systematic study of network curvature's theoretical properties under different models, including but not limited to SBM, is left as future work.

## C  Simulation

### C.1  Phase transition

As a measure of the performance of LRC preprocessing method, we conducted experiments testing recovery performance near exact recovery threshold. Phase transition in recovery performance has been widely used as a metric to evaluate the performance of numerous community detection algorithms (Abbe, 2018). Exact recovery threshold indicates theoretical conditions for SBM models where recovery of true community membership with probability $1 - o(1)$ is possible. The theorem for exact recovery in symmetric SBM (SSBM) is as follows.

**Theorem 1** *(Abbe et al., 2014; Mossel et al., 2015) Exact recovery in $SSBM(n, a\frac{\log n}{n}, b\frac{\log n}{n})$ is solvable and efficiently if $|\sqrt{a} - \sqrt{b}| > \sqrt{2}$ and unsolvable if $|\sqrt{a} - \sqrt{b}| < \sqrt{2}$.*

In the above theorem, $SSBM(n, a\frac{\log n}{n}, b\frac{\log n}{n})$ denotes a symmetric SBM with total $n$ nodes where $a\frac{\log n}{n}$ is within-community connection probability and $b\frac{\log n}{n}$ is across-community connection probability. For the experiment, we chose $n = 100$, $a$ ranging from 0 to 30 with increments 1, and $b$ ranging from 0 to 10 with increments 1. For each combination of a and b, we generated 100 SSBMs and computed the mean AMI or mean ARI. The theoretical exact recovery boundary, $\sqrt{a} - \sqrt{b} = \sqrt{2}$ is plotted in red for each plot for better comparison. For community detection algorithms, we chose spectral clustering and Leiden algorithms as representatives. The resulting heatmaps of ARI and AMI are provided in Figure 4 and Figure 5. Clearly, LRC preprocessing method improves the performance of both spectral clustering and Leiden algorithm, but still respect the theoretical recovery boundary.

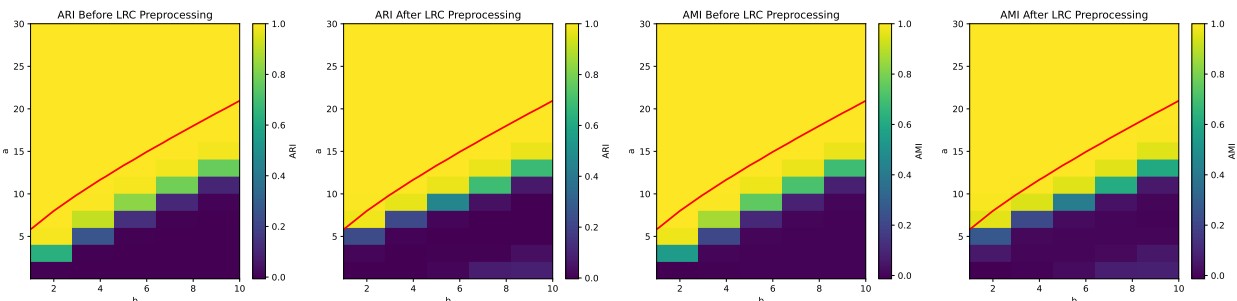

Figure 4: Phase transition in spectral clustering.

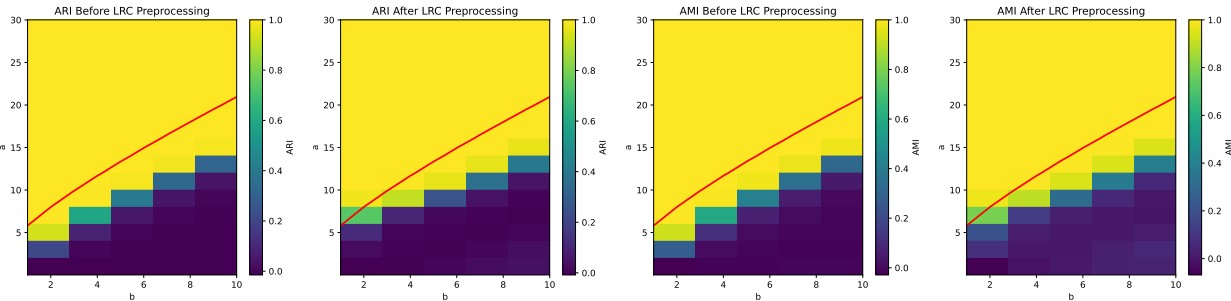

Figure 5: Phase transition in Leiden algorithm.

## C.2 Three additional scores

To access the effectiveness of LRC in community detection, we conducted simulations using SBM-generated networks. For a pair of edge probability within the community $p_1$ and edge probability across the community $p_2$ (with $p_2 < p_1$), we generate 100 replicates of the graph, each with $n = 100$ nodes. We evaluated three distinct scores, motivated by our proposed preprocessing method, to compare the performance of LRC against three other existing curvatures. The results are visually represented through heat maps, with the x-axis representing $p_1$, the y-axis representing $p_2$, and the color indicating the score.

**Proportion of Perfect Separation (PPS).** The first score we considered is the Proportion of Perfect Separation. For each graph replicate, we calculated the minimum curvature value among within-community edges and the maximum curvature value among across-community edges. A situation where the minimum within-community curvature exceeds the maximum across-community curvature indicates perfect separation of within and across community edges. This implies that our preprocessing would remove all across-community edges while retaining all within-community edges, enabling effective community detection by any reasonable method postprocessing. Mathematically, PPS is the proportion of networks satisfying $\inf_{(ij) \in E, z_i = z_j} \text{LRC}(ij) \geq \sup_{(ij) \in E, z_i \neq z_j} \text{LRC}(ij)$. The score ranges between 0 and 1, with higher values indicating better performance.

The diagonal heat maps in Figure 6 depict the extent of separation between within-community and across-community curvature distributions. A redder hue indicates a higher degree of separation. These maps suggest that all the four curvatures, including LRC, effectively differentiate community structures across a variety of $p_1, p_2$ pairs. The off-diagonal heat maps in the lower triangle compare the performance of each curvature with others (red for superior performance, blue for inferior), with a raw scale of $[-1, 1]$. The upper triangle heat maps also compare curvature performances, but with normalized ranges to amplify differences. Overall, LRC shows comparable performance in PPS compared to other curvatures.

**Average within-community Edge Removal Ratio (AER)** The second score, AER, provides a softer evaluation compared to PPS. While PPS focuses on perfect separation, AER quantifies the extent to which within-community edges might be incorrectly removed when aiming to eliminate all across-community edges. This score is particularly insightful, as it accounts for the potential drawback of our preprocessing method in mistakenly removing valuable within-community connections.

Mathematically, AER is defined as the ratio of the number of within-community edges, whose LRC values are smaller than the maximum LRC value of across-community edges, to the total number of within-community edges. In formula terms, AER is given by the proportion

$$\text{AER} := \frac{\left| \left\{ (ij) \in E, z_i = z_j : \text{LRC}(ij) < \sup_{(ij)] \in E, z_i \neq z_j} \text{LRC}(ij) \right\} \right|}{|\{(ij) \in E, z_i = z_j\}|}.$$

A score of 0 indicates ideal performance (no within-community edges are incorrectly removed), aligning with a PPS of 1. Conversely, an AER of 1 implies the extreme scenario where all edges are erroneously removed.

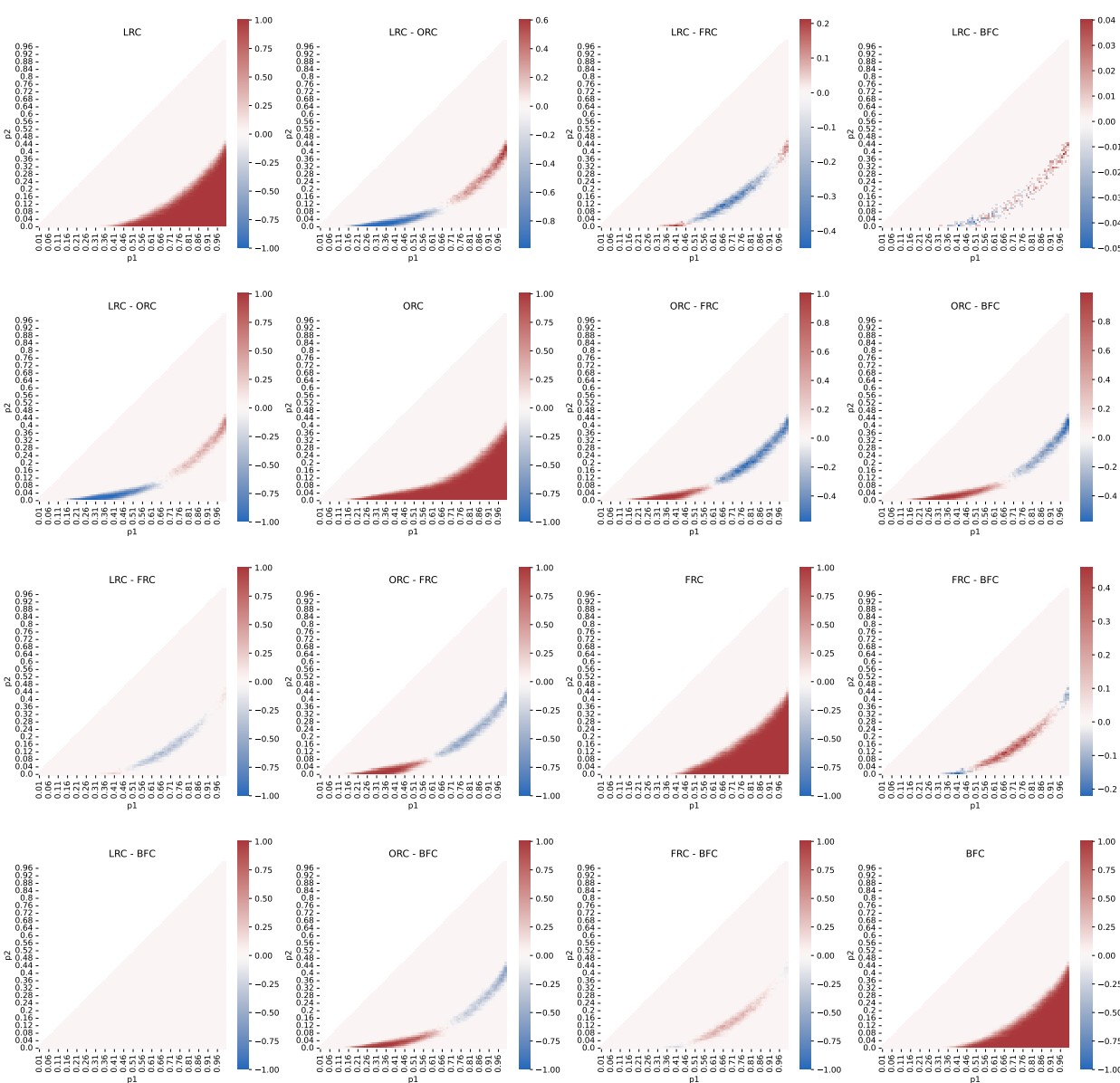

Figure 6: Comparison heat map for PPS.

Figure 7 presents the heat map of AER scores, organized similarly to the PPS heat map. The diagonal panels show the AER score for each curvature, while the off-diagonal panels compare the performance of different curvatures using the AER score. A visualization in these heat maps can provide insights into how effectively each curvature avoids the unintended removal of within-community edges, which is crucial for maintaining the integrity of the community structure during preprocessing.

**Average overlapping percentiles AOP.** The third score, AOP, is designed to quantify the extent of overlap between the curvature distributions of within-community and across-community edges in a more symmetric manner. This score captures the degree to which these two distributions intermingle, providing a nuanced view of the effectiveness of a curvature in distinguishing community structures.

Mathematically, AOP is calculated as follows: For each replicate graph, we determine the percentile of the minimum within-community LRC value within the distribution of across-community LRC values. We then

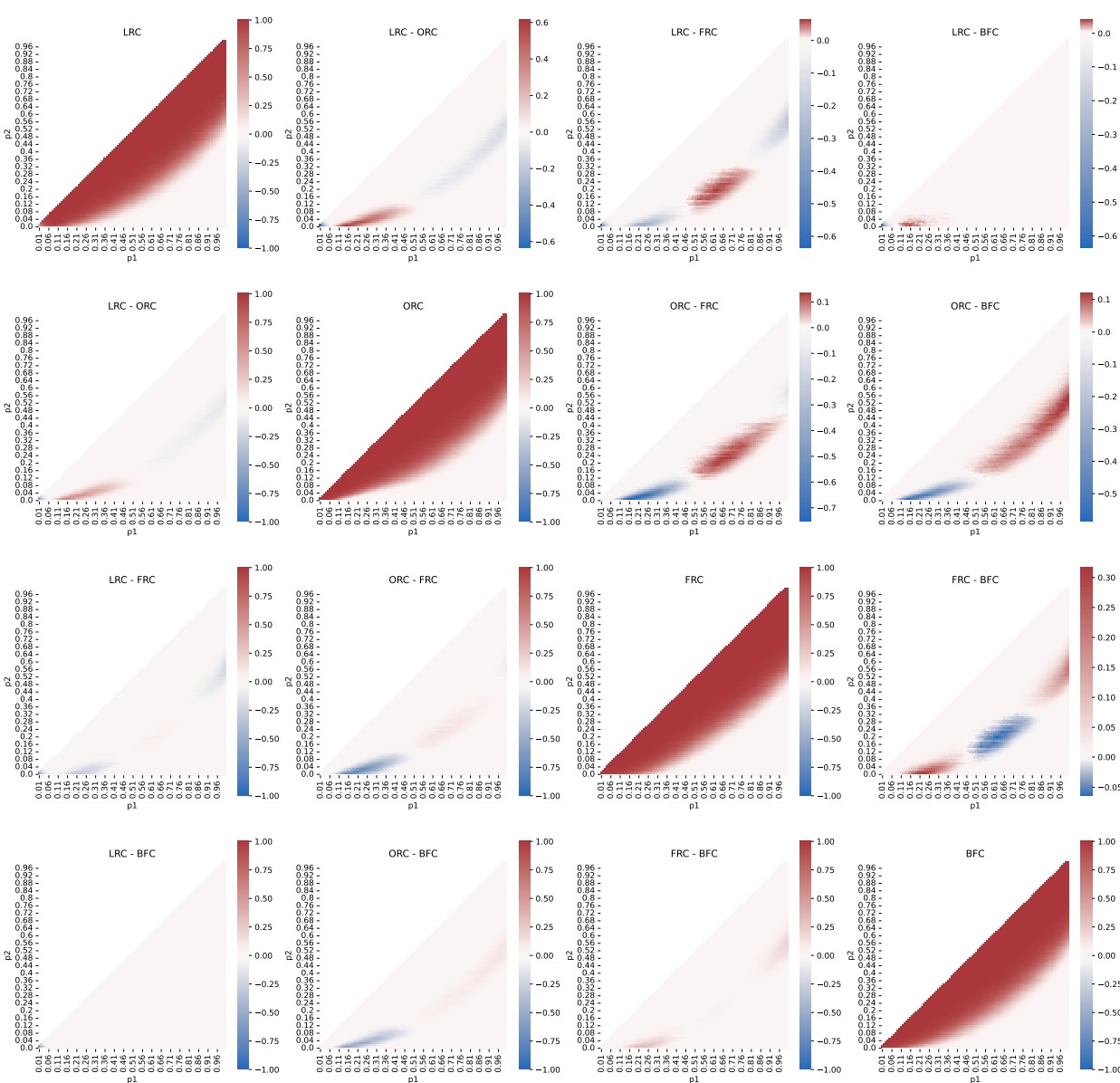

Figure 7: Comparison heat map for AER

calculate one minus the percentile of the maximum across-community LRC value within the distribution of within-community LRC values. The AOP score is the sum of these two quantities. Formally, it can be expressed as:

$$\text{AOP} := \mathscr{P}_{\inf_{(ij) \in E, z_i = z_j}} \left( \{\text{LRC}(ij) : (ij) \in E, z_i \neq z_j\} \right) + 1 - \mathscr{P}_{\sup_{(ij) \in E, z_i \neq z_j}} \left( \{\text{LRC}(ij) : (ij) \in E, z_i = z_j\} \right),$$

where $\mathscr{P}_a A$ is the $a$-th percentile of set A.

In the ideal scenario where there is no overlap, the first percentile would be 1 (indicating the minimum within-community LRC is at the highest end of the across-community distribution), and the second percentile would also be 0 (indicating the maximum across-community LRC is at the lowest end of the within-community distribution), resulting in an AOP score of 2. Conversely, in the worst-case scenario where there is complete overlap, the AOP score becomes 0. Figure 8 displays the heat map visualization of AOP scores, arranged

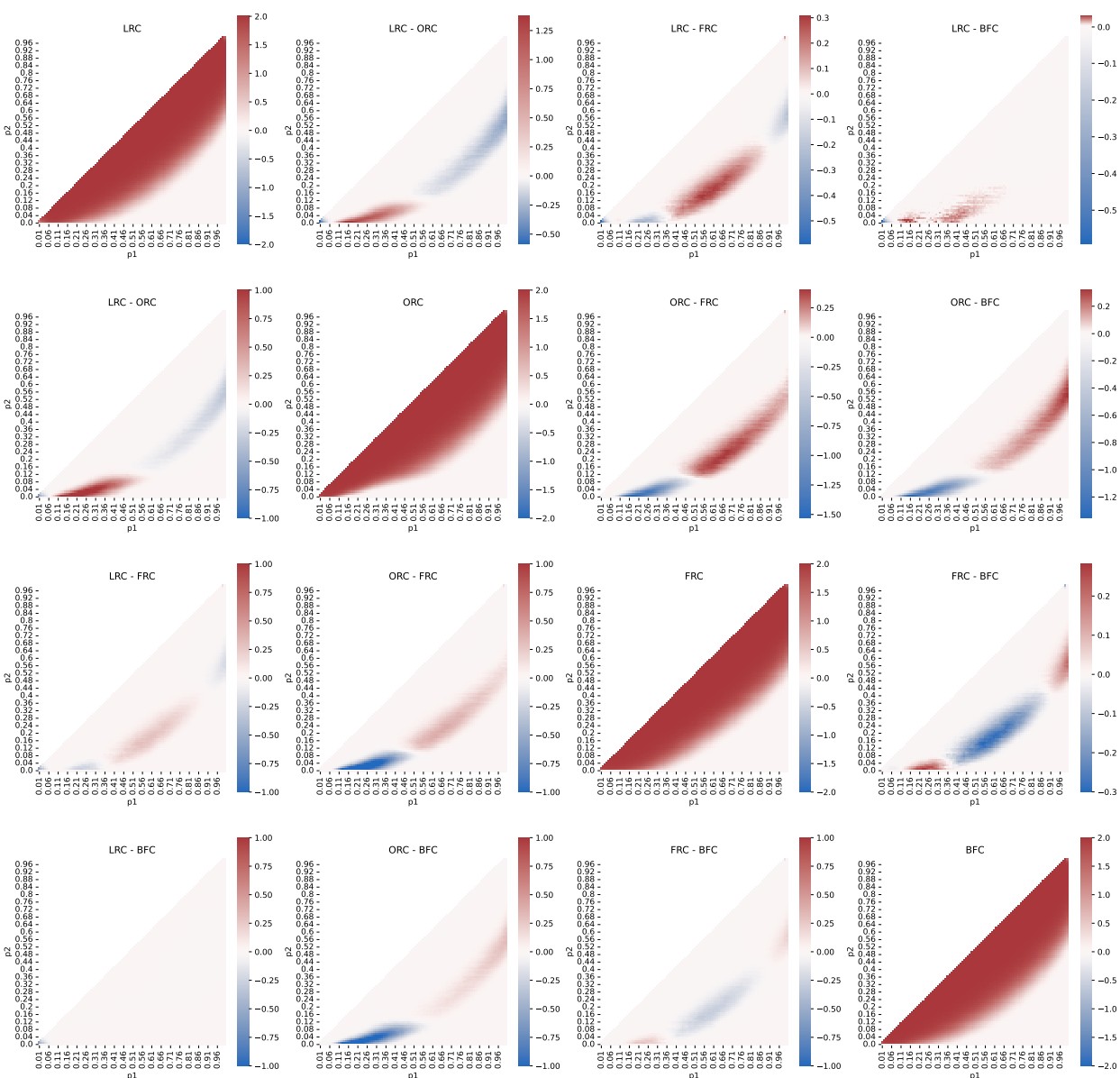

Figure 8: Comparison heat map for AOP.

similarly to the previous scores. The diagonal panels show the AOP for each curvature, while the off-diagonal panels compare different curvatures using the AOP measure. This visualization aids in understanding the extent to which each curvature can differentiate community structures by evaluating the overlapping of curvature distributions.

In summarizing the evaluations conducted using PPS, AER, and AOP, we observe that the four curvatures – LRC, FRC, BFC, and ORC – exhibit comparable performance in community detection. None of the curvatures consistently outperforms the others across all metrics and all pairs of $p_1, p_2$, indicating a balanced landscape of effectiveness.

However, when considering computational efficiency, LRC and FRC emerge as the fastest, both offering $O(n)$ complexity. The crucial difference is that FRC is not scale-free, but LRC boasts this advantageous property, making it particularly suitable for analyzing large-scale networks where scalability is key. This distinction

positions LRC as an ideal candidate for our proposed preprocessing method. Nevertheless, it is important to note that if specific scenarios or requirements strongly favor other curvatures, our preprocessing approach remains adaptable and can be effectively applied in a broader context.

## D   Comparison with network denoising algorithms

Network denoising is a process that aims to remove erroneous or noisy edges to reveal its underlying structure(Luo et al., 2021; Yu et al., 2021). However, the LRC preprocessing step is not intended to treat bridges (edges with low curvature) as noise. Instead, we are identifying edges that has informative signals that, when removed, improve the performance of community detection by making community structures more distinct. To compare denoising method with our preprocessing method, we implemented the NR method by Yu et al. (2021) and applied it to the NCAA football network dataset. The results are included in the Table 6. We observe that the AMI and ARI scores after NR denoising are lower than those obtained without any preprocessing for all four community detection methods considered. This suggests that while NR is a general-purpose denoising method, our LRC-based preprocessing is specifically tailored to enhance community detection performance.

Table 6: Community detection algorithms evaluation for NCAA Football League network

| Scores | Algorithms | | | |
| --- | --- | --- | --- | --- |
| | Label Propagation | Leiden | Girvan–Newman | Walktrap |
| ARI: before | 0.75 | 0.81 | 0.84 | 0.82 |
| ARI: after LRC | 0.90 | 0.90 | 0.84 | 0.90 |
| ARI: after NR | 0.40 | 0.47 | 0.00 | 0.57 |
| AMI: before | 0.83 | 0.86 | 0.87 | 0.86 |
| AMI: after LRC | 0.90 | 0.90 | 0.88 | 0.90 |
| AMI: after NR | 0.73 | 0.67 | 0.01 | 0.76 |

## E   Additional experimental details

### E.1   Hyperparameters for community detection

All algorithms implemented in this paper are from Python package CDlib (Rossetti et al., 2019). The hyperparameters are as follows:

1. NCAA Football League network

   **Label Propagation:** NA.

   **Leiden:** Initial membership = None, weights= None.

   **Girvan-Newman:** Level = 10.

   **Walktrap:** NA.

2. DBLP collaboration network

   **Angel:** Threshold = 0.5, minimum community size = 3.

   **Ego-networks:** Level = 1.

   **K-clique:** $K = 3$.

   **SLPA:** $t = 20, r = 0.1$.

3. Amazon product co-purchasing network

   **Angel:** Threshold = 0.5, minimum community size = 3.

**Ego-networks:** Level $= 1$.

**K-clique:** $K = 3$.

**SLPA:** $t = 20, r = 0.1$.

4. YouTube social network

**Angel** Threshold $= 0.5$, minimum community size $= 3$.

**Ego-networks** Level $= 1$.

**K-clique** $K = 5$.

**SLPA** $t = 20, r = 0.1$.

## E.2 Code and data availability

All codes can be found in `https://github.com/parkyunjin/LowerRicciCurv`

The four real datasets used in this paper can be downloaded in the following websites:

1. NCAA Football League network: `https://websites.umich.edu/~mejn/netdata/` under "American College football". The graph is provided in the Graph Modeling Language (.gml) format.

2. DBLP collaboration network: `https://snap.stanford.edu/data/com-DBLP.html` This graph is represented as NetworkX objects, provided by the CDlib Python package.

3. Amazon product co-purchasing network: `https://snap.stanford.edu/data/com-Amazon.html` This graph is represented as NetworkX objects, provided by the CDlib Python package.

4. YouTube social network: `https://snap.stanford.edu/data/com-Youtube.html` This graph is represented as NetworkX objects, provided by the CDlib Python package.

