# OpenReview forum: "Lower Ricci Curvature for Efficient Community Detection"
_TMLR — Accepted by TMLR_

### Review · Reviewer_tfRj · 2024-11-26

**Summary Of Contributions:**

This submission proposes a new notion of "discrete curvature" for graphs, called *Lower Ricci Curvature* (LRC), with the goal of improving community detection algorithms. There are a few other methods given by prior work. This one aims to keep the best aspects of all of them, as it is computationally lightweight and, perhaps, easier to interpret.

In the main body of the submission, we get an overview of prior work, a definition and discussion on the new curvature notion, and experiments. There are some examples on toy models. The main experiments evaluate LRC as a pre-processing step: if one computes LRC for every edge and then removes edges with low values, does it improve the downstream performance of existing algorithms for community detection?

**Audience:**

Yes

**Broader Impact Concerns:**

None.

**Claims And Evidence:**

Yes

**Requested Changes:**

Once these changes are made I will happily recommend acceptance. Items 3, 4, and 5 are small edits.
1. If "scale-free" means something precise, include a definition and prove that LRC satisfies it. If it does not have a precise definition, please point out that you use it informally and spend some more time discussing what it means.
1. Address my previous comments on computation and other algorithms.
1. Typo before Def 1, it should say "FRC."
1. Top of p5, Ollivier's happened simultaneously with what?
1. The plots, like Figs 1 and 2, would be easier to read with bigger text.

**Strengths And Weaknesses:**

This submission will be of interest to some in the TMLR audience. The paper with written with care and it is easy to follow the points the authors make. All the claims seem supported, modulo some details I address below.

The paper should better address (i) why these notions of discrete curvature are analogs of the continuous notion and (ii) why discrete curvature is useful in community detection. Currently, these points are rushed over or left implicit. (Such a discussion doesn't need to be very formal.)

I don't understand what is meant by the claim that LRC is "scale-free," which is an issue as this is one of the central claimed contributions.

I found the submission's discussion of computation to be lacking. Here are a few comments.
1. How is the graph presented?
1. I believe algorithms based on fast matrix multiplication, such as that of Itai & Rodeh, allow us to compute all-pairs FRC in time faster than $O(mn)$ when the graph is dense. It's worth mentioning these, especially around statements like "the computational cost for calculating FRC...", which might be better phrased as "in our setting, the best-known algorithm...".
1. There are $\tilde{O}(n)$ algorithms for community detection in SBMs under certain parameter regimes, e.g. [1]. I would like to see a brief discussion on this line of work and whether or not it's relevant.
1. I don't understand why we switch from big-Oh notation in Table 2 to clock time in Table 3. The sentence containing "due to the complex nature of the community detection... computational complexity is not reported here." did not clarify things.

[1] Wang, Peng, Zirui Zhou, and Anthony Man-Cho So. "A nearly-linear time algorithm for exact community recovery in stochastic block model." International Conference on Machine Learning. PMLR, 2020.

---

> ### Author Response · Authors · 2025-01-23
>
> We thank the reviewer for the valuable feedback and constructive comments, which have greatly helped improve our manuscript. We address each point in detail below and have revised the paper accordingly, marked in blue.
>
> **Regarding "scale-free"**
>
> We appreciate the reviewer’s feedback and the opportunity to clarify the term "scale-free" as it pertains to our work. In this context, "scale-free" means that the range of curvature values is independent of the network size (e.g., the total number of nodes or edges). This property is crucial for comparing curvature values across networks of different scales and ensures that the curvature is robust to variations in network size and structure.
>
> To elaborate, FRC is not scale-free because it directly depends on the overall size of the network. As a result, it is challenging to compare curvature values across networks of different sizes. In contrast, ORC, BFC, and LRC are bounded measures, regardless of network size. This boundedness enables these curvatures to be interpreted consistently across networks of varying sizes.
>
> The scale-free property is particularly important in applications such as distinguishing between cancer and non-cancer networks (e.g., Sandhu et al., 2015), where the ability to compare curvature values across networks is essential for identifying meaningful patterns.
>
> We have revised the manuscript accordingly.
>
> **The analogs of continuous curvature**
>
> We appreciate the reviewer’s comment and have clarified these points in the revised manuscript.
>
> Discrete curvatures are analogs of continuous Ricci curvature because, under specific conditions, they converge to their continuous counterparts as the graph becomes dense and resembles a Riemannian manifold. For ORC, this is discussed in van der Hoorn et al. (2021), and for FRC, in Sreejith et al. (2016). These results establish a theoretical bridge between the discrete and continuous notions of curvature.
>
> Discrete curvature is useful in community detection because it effectively distinguishes inter-community edges (typically with lower curvature) from intra-community edges (typically with higher curvature). This distinction is evident in Figure 1 of the introduction, where edges connecting different communities exhibit lower ORC values compared to edges within the same community.
>
>
> **How is the graph presented?**
>
> The graphs used in our experiments are represented as follows:
>
> NCAA Football League Network: The graph is provided in .gml format, which explicitly lists the nodes, edges, and associated community labels. This format is commonly used for small to medium-sized networks and is compatible with various graph-processing tools.
>
> DBLP Collaboration Network, Amazon Product Co-purchasing Network, and YouTube Social Network: These larger-scale graphs are represented as NetworkX objects, provided by the CDlib Python package. A NetworkX graph object stores nodes and edges as dictionaries, where each node is associated with its metadata (e.g., community labels in our case). This representation is efficient for sparse graphs and well-suited for the operations needed in our analysis.
>
> These are the formats in which the data were obtained, and we worked with them directly in our analysis. We have updated appendix E.2 accordingly.
>
>
> **Computational cost of the curvatures**
>
> We appreciate the reviewer pointing out the relevance of fast matrix multiplication-based algorithms for dense graphs. To avoid confusion, we have revised our discussion to focus on the computational cost of calculating the curvature for a single edge, rather than for the entire graph. This change removes the dependency on the number of edges (m) and makes the complexity clearer across different settings, as shown in the updated Table 1.
>
> **Nearly-linear time algorithm**
>
> Thank you for bringing this work to our attention. In fact, our LRC-based preprocessing method can be combined with any community detection algorithm, and using an efficient method like the one proposed by Wang et al. (2020) would be a valuable enhancement, making the entire approach even more efficient. We have added a brief discussion of this work and cited the referenced paper in the revised manuscript.
>
> **Computation complexity**
>
> We appreciate the reviewer’s comment and agree that the explanation regarding the use of clock time instead of Big-O notation could be clearer. For the methods used in Table 3 (e.g., Angel, Ego, K-clique, and SLPA), explicit time complexity analyses are not consistently available in the literature, and we used their existing implementations as provided by widely used packages. Instead, we reported empirical clock times to provide a practical comparison of their performance in our experiments. We have clarified this in the revised manuscript.
>
> **Typos**
>
> Thanks for catching the typos, we have fixed them in the revised manuscript.

---

> > ### Comment · Reviewer_tfRj · 2025-02-14
> >
> > Thank you for your detailed response and revisions, they completely resolve the (minor) issues I had with the first version.
> >
> > (Apologies for my slow reply.)

---

### Review · Reviewer_N1VX · 2024-12-09

**Summary Of Contributions:**

In this article, the authors propose a new type of measure for quantifying the curvature in networks, called the Lower Ricci curvature (LRC). Compared to other measures of curvatures, such as the Ollivier-Ricci curvature (ORC), the Forman-Ricci curvature (FRC), or its balanced version (BFC), the LRC enjoy several good properties: it captures well curvature information in the context of community detection (as demonstrated by its theoretical links with the diameter, Cheeger constant, and graph Laplacian spectrum), while being much faster to compute than its competitors (as its complexity is linear in the number of graph nodes and edges), making it highly scalable. Empirical evidence is demonstrated on large networks, where it is shown that an LRC-based edge pruning strategy results in significant improvements for most community detection methods, while keeping a small running time.

**Audience:**

Yes

**Claims And Evidence:**

Yes

**Requested Changes:**

---I would have appreciated a more intuitive description of the formulas associated to the FRC, BFC, ORC and LRC. Otherwise, the equations provided in the various definitions are a bit cryptic.

---In terms of baselines in the experiment, why not compare against the non-iterative preprocessing versions of the FRC, BFC and/or ORC? As far as I understand, Algorithm 1 could be used with any of the other curvature measures in a non-iterative way, making them more scalable. This could maybe allow to even include ORC in the baselines for the various experiments. Alternatively, would an iterative version of Algorithm 1 with LRC perform even better?

---I am a bit puzzled about some of the results presented in Table 5. Do the authors have an explanation for the huge increase in running time when using SLPA after FRC preprocessing? Also, given that the running times of K-clique are exactly the same after FRC and LRC preprocessings, I would have expected that the simplified networks are the same, resulting in the same F1 scores as well. However the F1 scores are different. Any idea of what is happening?

Typos:

---definition of $\Delta(i,j)$ above Proposition 1: I think $s_i$ and $s_j$ should be $s_{i,j}$ and $s_{j,i}$ to be consistent with Definition 2.

---appendix A: "$\Delta\geq .$" ---> $0$ is missing.

**Strengths And Weaknesses:**

Overall, I am quite positive about this work: the problem being studied is quite important as meaningful graph curvature measures are known to not scale well, the text is very well written and very clear, and the numerical experiments are convincing. I just have a few questions/remarks that I would like the authors to address (see below).

---

> ### Author Response · Authors · 2025-01-23
>
> We thank the reviewer for the thoughtful comments and suggestions, which have helped improve the clarity and rigor of our work. We address each comment below and have revised the manuscript accordingly, marking the changes in red.
>
>
> **Intuitive descriptions of the formulae**
>
> Below, we provide simplified descriptions of each curvature, along with their conceptual underpinnings:
>
> ORC: ORC quantifies the change in probability distributions across the neighborhoods of two nodes connected by an edge. It is inspired by optimal transport theory, where the curvature measures how close the local connectivity structure is to that of a flat graph. A larger ORC value indicates stronger cohesion between the two neighborhoods, while a smaller or negative value suggests a bridge-like connection.
>
> FRC: FRC evaluates the "structural support" of an edge by considering both the degrees of the nodes it connects and the number of triangles (shared neighbors) that the edge participates in. The intuition is that edges with more shared neighbors and lower-degree endpoints are more ``integral" to the network's structure, leading to higher FRC values.
>
> BFC: BFC refines FRC by addressing its scale dependence and skewness toward negative values. BFC incorporates additional terms that normalize the contribution of shared neighbors and node degrees, making it bounded and scale-free. This modification ensures that the curvature captures essential topological features, such as bottlenecks, more consistently.
>
> LRC (our proposed curvature): LRC is a simplified, computationally efficient approximation of BFC, designed to retain its community-detection properties while reducing computational overhead. It focuses on the shared neighbor contributions relative to the node degrees, offering a practical curvature measure that distinguishes within-community edges (higher LRC) from inter-community edges (lower LRC).
>
> We also note that BFC and ORC have theoretical connections to continuous Ricci curvature when the network is a dense sample of a manifold, as discussed in Topping et al. (2021). Additionally, LRC was motivated by computational efficiency while maintaining essential theoretical properties (Proposition 1 in the paper) and practical utility in community detection.
>
> For further details on the theoretical foundations and derivations of these curvature measures, see also Forman (2003), Topping et al. (2021), and Ollivier (2007). The derivation and motivation for LRC are elaborated upon in our manuscript (Section 3 and Proposition 1), while the intuition of existing curvatures are added to Section 2.
>
>
> **Baselines**
>
> Algorithm 1 can indeed be applied with any curvature measure in a non-iterative way. We have already compared the non-iterative preprocessing version of LRC with FRC and BFC in the application section (see Tables 2–5).
>
> We did not include ORC in these experiments, not because of its iterative nature, but due to its computational cost. Calculating ORC of a single edge alone is $O(n^3)$, which makes it impractical for large datasets like those used in our paper, Even in a non-iterative manner.
>
> Regarding an iterative version of Algorithm 1 with LRC: This approach would involve calculating the LRC of all edges in each iteration, increasing the computational complexity to $O(nm^2)$ in the worse case. While this might provide slightly better results, the computation time and resource requirements would grow significantly, especially for large or dense networks. Since our primary focus is computational efficiency, we prioritized methods that balance performance and scalability.
>
> **Table 5**
>
> We thank the reviewer for pointing this out. The discrepancies in Table 5 were due to a typo, which has been corrected in the revised manuscript.
>
> **Typos**
>
> Thanks for catching the typos. We have fixed them in the revised manuscript.

---

> > ### Comment · Reviewer_N1VX · 2025-01-31
> >
> > Thank you for your comments.

---

### Review · Reviewer_GeSG · 2025-01-10

**Summary Of Contributions:**

This paper introduces a measure of ``curvature'' on graphs that is less computationally intensive than previously used notions of curvature.  It then uses this in a preprocessing algorithm to improve the accuracy of existing community detection methods.  Experiments are performed on stochastic block models as well as various real networks with known ground-truth communities.  Connections are drawn to other graph structural parameters, such as the Laplacian spectral gap and the diameter.

**Audience:**

Yes

**Claims And Evidence:**

Yes

**Requested Changes:**

1.) I feel that addressing weaknesses 1-4 is important for strengthening the paper.  However, a faithful reading of the acceptance criteria for this journal leads me to conclude that I cannot justify rejection the basis of these weaknesses.

Regarding weakness 5, I am not really certain that it can be fixed, but perhaps the authors can comment on it.

**Strengths And Weaknesses:**

Strengths:

1.) The proposed notion of ``curvature'' is more efficiently computable than previously introduced notions.

2.) The proposed notion of curvature is related to graph structural properties, such as the Laplacian spectral gap, that are associated with cut sizes in graphs.

3.) For several example datasets, either running time, accuracy, or both are improved by the use of LRC.

Weaknesses:

1.) The theoretical analysis seems to be a little trivial.  Furthermore, it misses some low-hanging fruit: it is very easy to give a high-probability asymptotic expression for the lower Ricci curvature of edges in the stochastic block model, under the regime that is covered by experiments.  This can be used to show that with probability $1-o(1)$, ground-truth community memberships can be recovered exactly.  I feel that this direction should be pursued for completeness.  The crux of the analysis is to first calculate the expected value of $n_{i,j}$ and $n_i$, then show that these are both well-concentrated around their expected values (using a Chernoff bound).  A union bound then allows one to conclude that with probability $1-o(1)$, $n_{i,j}$ and $n_i$ are uniformly close to their expectations (more specifically, $n_{i,j} = \mathbb{E}[n_{i,j}]\cdot (1 + o(1))$ and $n_i = \mathbb{E}[n_i] \cdot (1 + o(1))$), for all vertices $i, j$.  The remainder of the analysis plugs these approximations into the definition of the LRC and then simplifies.

2.) The experiments on stochastic block models do not cover challenging cases of community detection near the exact recovery threshold.  See https://www.jmlr.org/papers/volume18/16-480/16-480.pdf and surrounding works for details.  Indeed, away from this threshold, rather trivial community detection algorithms recover ground-truth community memberships with probability $1-o(1)$.  I feel that the authors should cite this and surrounding work and should additionally perform experiments using SBMs with parameters near the exact recovery threshold.

3.) It would be useful to see experiments using alternative definitions of curvature -- for example, the authors mention $\frac{n_{i,j}}{\max \{n_i,n_j\}}$.  The reason that I think this would be useful is that one can use this alternative notion to perform exact community recovery in the SBM in the regime that the authors have empirically analyzed.  Thus, I fear that the proposed curvature notion is rather arbitrary.

4.) Essentially, calculation of LRCs can itself be viewed as the basis of a community detection algorithm that partitions edges into between-community and within-community subsets, with an implicit confidence measure.  The preprocessing step simply discards the low-confidence edges.  One could view this as a denoising procedure (which is not really necessary in the SBM case for the parameter regime that the authors study).  If this is the case, then I feel that there may be some relevant work to which the authors should compare (e.g., https://www.sciencedirect.com/science/article/abs/pii/S0378437123002364).

5.) It seems like the wall clock running time results for some datasets are not very good.  For instance, the running times in Table 4 seem to indicate that for three out of the four considered algorithms, the use of LRC significantly increased running time compared to the use of FRC and even, in two cases, compared to no preprocessing.

---

> ### Author Response · Authors · 2025-01-23
>
> We thank the reviewer for the constructive feedback and insightful comments, which have helped improve our manuscript. We address each point in detail below and have revised the paper accordingly, with changes highlighted in orange.
>
> **Concentration theory**
>
> Thanks for the insightful comment. To address your concern, we added a new section in the appendix where we discuss the theoretical bounds for $n_{ij}$ under SBM, as a first step towards understanding the theoretical performance of LRC. Specifically, we calculate the expected values of $n_{ij}$ for within-community and across-community edges, apply Chernoff bounds, and provide a union-bound-based estimate for the probability of exact recovery of community memberships.
>
> However, these calculations are specific to $n_{ij}$ and do not fully extend to LRC, which involves additional dependencies on $n_i$ and $n_j$. Rigorous analysis of LRC under SBM requires further investigation, as it introduces additional complexity. Due to these challenges and the page limit, we have added a note in the discussion section to highlight this as an important direction for future work. A systematic study of network curvature's theoretical properties under different models, including but not limited to SBM, is left as future work.
>
>
> **Exact recovery threshold**
>
> Thank you for the insightful suggestion regarding experiments near the exact recovery threshold for SBMs. We have conducted additional experiments using SBMs with $n$ nodes from two communities, where the within-community edge probability is $a \frac{\log n}{n}$ and the across-community edge probability is $b\frac{\log n}{n}$. We ran the Leiden algorithm and spectral clustering for various combinations of $a$ and $b$ over 100 replications. The resulting average ARI and AMI have been included as the heatmap in the revised manuscript, Appendix C.1.
>
> We also plotted the theoretical exact recovery boundary, $\sqrt{a} - \sqrt{b} = \sqrt{2}$, in red, and found that it holds as expected: exact recovery is only possible for combinations of $a$ and $b$ above this boundary. Importantly, ARI and AMI were consistently improved by the LRC preprocessing step. This further demonstrates the effectiveness of our proposed method.
>
> Additionally, we have cited relevant references, including the work you suggested, to provide context for the exact recovery boundary in SBMs.
>
>
>
> **Alternative definitions**
> We acknowledge that our definition of curvature is not the only possible way to define a reasonable measure. However, we believe LRC is at least a reasonable definition, as supported by Theorem 1, Corollary 1 and the numerical results in our paper.
>
> Regarding the alternative curvature measure $\frac{n_{ij}}{\max(n_i, n_j)}$, there are several reasons we chose not to use it. First, this measure is always positive, which conflicts with the intuition that Ricci curvature can be positive, negative, or zero, reflecting different geometric properties: positive for sphere-like structures, negative for hyperbolic spaces, and zero for flat spaces. Second, this measure is an upper bound of ORC and FRC, meaning Corollary 1 would no longer hold. This weakens the connection between the curvature and the Cheeger constant, a key quantity for understanding the network. Third, it may fail to distinguish between different structural situations. For example, if two nodes $i,j$ share no neighbors, this curvature would always be zero, regardless of other factors.
>
> To illustrate, consider a graph composed of two fully connected subgraphs, each with $n$ nodes, connected by a single edge $e$. Then, $\mathrm{LRC}(e)\xrightarrow[]{n\to\infty}-2$ reflecting its role as a “bridge” between large subgraphs. In contrast, the alternative curvature would remain zero, regardless of $n$. This fails to capture the intuition that the bridge becomes more significant as the fully connected subgraphs grow larger. Furthermore, for a straight-line graph (path graph), the alternative curvature would also be zero for any edge, as there are no common neighbors. As a result, this measure cannot distinguish between a bridge and a straight line, failing to reflect important structural differences.
>
> While this alternative measure might perform well in specific contexts, the drawbacks mentioned above led us to not adopt it in practice. We agree that exploring alternative curvature definitions is an interesting direction for future work. In response to the reviewer’s suggestion, we have added a brief discussion on this in the revised discussion section.

---

> > ### Author Response · Authors · 2025-01-23
> >
> > **Denoising comparison**
> >
> > We thank the reviewer for this observation and the suggested reference. To clarify, our LRC-based preprocessing is not exactly a denoising method. Below, we outline three key points of distinction:
> >
> > 1. The LRC preprocessing step is not intended to treat bridges (edges with low curvature) as noise. Instead, we treat these edges as meaningful signals that, when removed, improve the performance of community detection by making community structures more distinct.
> >
> > 2. The method suggested by the reviewer (referred to as NR) does not remove edges but instead updates the weights of all edges. This makes it fundamentally different from our approach, which simplifies the graph by explicitly removing certain edges. Thus, a direct comparison may not fully capture the differences in their intended purposes.
> >
> > 3. In response to the reviewer’s suggestion, we implemented the NR method and applied it to the NCAA football network dataset used in our manuscript. The results are included in Table 6 in Appendix D. We observe that the AMI and ARI scores after NR denoising are lower than those obtained without any preprocessing for all four community detection methods considered. This suggests that while NR is a general-purpose denoising method, our LRC-based preprocessing is specifically tailored to enhance community detection performance.
> >
> > **Clock running time**
> >
> > We appreciate the reviewer’s observation regarding the wall clock running times in Table 4. To clarify, the reported times are the sum of the LRC preprocessing step and the running time of the community detection method itself, which is determined by the specific algorithm and is beyond our control. Our focus is to show that the LRC-based preprocessing method can improve existing community detection methods, and that the preprocessing step itself is computationally very efficient due to the scalability of LRC.
> >
> > The additional preprocessing step using LRC may increase the total running time in some cases if the reduction in network size does not sufficiently offset the cost of preprocessing. Conversely, it can decrease the overall runtime by simplifying the network. However, we do not claim that the total runtime with LRC preprocessing will always be faster than with FRC preprocessing or without preprocessing.
> >
> > It is worth noting that BFC and ORC are so computationally expensive that they are not feasible for the large-scale networks considered in our paper, which underscores the efficiency of LRC. We have clarified these points in the revised Section 5.

---

### Decision · Action_Editor_Lgc5 · 2025-02-17

**Recommendation:** Accept as is

**Comment:**

I thank the authors for addressing the comments from all three reviewers and revising their paper accordingly. Despite the lack of thorough theoretical analysis and justification, this work does demonstrate merits of the proposed approach to improving existing community detection algorithms, at least empirically. With the revisions, all reviewers are supportive of acceptance of this paper for publication in TMLR.

**Audience:**

Community detection in general and techniques for improving existing algorithms in particular are of interest to some readers in the TMLR audience.

**Claims And Evidence:**

The claims made in the submission are mainly supported through empirical studies. While conducting more thorough theoretical analysis would be a good direction to pursue, the reviewers do not consider its inclusion to be necessary for the acceptance of this work. I agree with them on this.